# Self-assembly of the smallest and tightest molecular trefoil knot

Zhiwen Li[1,2], Jingjing Zhang[1,2], Gao Li [1,2] ✉ & Richard J. Puddephatt [3] ✉

Molecular knots, whose synthesis presents many challenges, can play important roles in protein structure and function as well as in useful molecular materials, whose properties depend on the size of the knotted structure. Here we report the synthesis by self-assembly of molecular trefoil metallaknot with formula $[Au_6\{1,2\text{-}C_6H_4(OCH_2CC)_2\}_3\{Ph_2P(CH_2)_4PPh_2\}_3]$, **Au₆**, from three units of each of the components $1,2\text{-}C_6H_4(OCH_2CCAu)_2$ and $Ph_2P(CH_2)_4PPh_2$. Structure determination by X-ray diffraction revealed that the chiral trefoil knot contains only 54 atoms in the backbone, so that **Au₆** is the smallest and tightest molecular trefoil knot known to date.

Knots have played key roles in areas ranging from biological evolution to the advancement of human society. In biology, knot structures with DNA[1], RNA[2], and protein[3] units are important in living beings. Furthermore, the development of human society relied on the ability to use knots for making useful tools and robust protective shelters. Nowadays, many people pray for happiness with red Chinese knots, which are the most complex knots made by hand. The intrinsic beauty and usefulness of the knot has stimulated a wave of research on knots in various scientific fields, including mathematics, physics, biology, and chemistry.

Mathematicians and physicists have predicted an infinite number of primary knots of increasing complexity, but the synthesis by chemists of molecular knots remains challenging. Knots are classified according to the minimum number of crossings when the reduced form of the structure is projected onto a two-dimensional surface, with a subscript indicating the isomer number in the standard table of knots. The simplest knot is the trefoil knot $3_1$, the only isomer with three crossings, while one of the most complex molecular knots is $8_{19}$, with 8 crossings. The number of potential isomers increases with complexity and the synthesis becomes more challenging. The most common synthetic procedure is to prepare carefully designed helical chains, to orient them by coordination to one or more metal ion templating agents, to close the rings by bridging the chain ends, and finally to remove the metal ions to give the desired molecular knot. In 1989, Sauvage and co-workers pioneered this method for synthesis of the first molecular trefoil knots[4,5]. The impressive advances since then have been the subject of regular excellent reviews[6–12]. Several new

synthetic procedures have been developed, using different template methods to align the chains or simply by using self-assembly, often in the synthesis of metallaknots[6,12–17].

Much of the recent research on molecular knots has been concentrated on the synthesis, structure, properties and applications of larger more complex examples, but there is also a keen interest in synthesis of the smallest, tightest knots[6,9]. There is no absolute way to estimate the tightness of a knot. However, for a given knot type, they can be ranked according to the number of atoms in the shortest path along the knotted strand or, more generally for all knots, by this number divided by the number of crossings (the backbone crossing ratio or BCR)[9]. A smaller value of chain length or BCR corresponds to a tighter knot. For organic trefoil knots, the minimum reported chain length is 76 and the number of crossings is three, so the BCR = 25.3[6,18]. The smallest trefoil metallaknot formed by self-assembly has a 69-atom chain length with BCR = 23[12]. Most organic molecular knots have BCR in the range 27–33, and the tightest is the $8_{19}$ knot, reported by Leigh in 2017[19], which has a 192 atom backbone and BCR = 24. For trefoil knots based on a polymethylene chain, $(CH_2)_n$, for which no strong secondary bonding forces are expected, early predictions based on space-filling models suggested that a minimum ring size of $n = 45$–50 atoms might be possible[9,20], but more recent quantum chemical calculations suggest that the trefoil knot would be more stable than the simple ring only when the ring size is at least 59 (BCR = 19.7)[21]. Most synthetic procedures for molecular knots require the introduction of functional groups to take part in the templating step, so they are not easily optimized for making the smallest

[1]State Key Laboratory of Catalysis, Dalian Institute of Chemical Physics, Chinese Academy of Sciences, Dalian 116023, China. [2]University of Chinese Academy of Sciences, Beijing 100049, China. [3]Department of Chemistry, University of Western Ontario, London N6A 5B7, Canada. ✉e-mail: gaoli@dicp.ac.cn; pudd@uwo.ca

knots. Hence, how small a molecular knot could be remains an open question[6,9].

This article describes the synthesis of a trefoil $Au_6$ metallaknot ($Au_6$) with only 54 atoms in the backbone (BCR = 18). It was prepared simply by self-assembly of units of a digold(I) diacetylide and a diphosphine ligand, a method previously reported to yield only gold catenane structures[22]. Characterization by single crystal X-ray diffraction provides convincing evidence that $Au_6$ represents both the smallest and the tightest molecular knot known to date.

## Results

Many alkynyl gold(I) complexes $(RCCAu)_n$ are known and form oligomeric or polymeric structures in which the alkynyl group bridges between gold(I) centers via combinations of σ- and π-bonds[23–26]. The complex 1,2-$C_6H_4(OCH_2CCAu)_2$, **L-Au₂** (Fig. 1), was prepared by reaction of 1,2-$C_6H_4(OCH_2CCH)_2$, [AuCl(SMe₂)] and base according to our standard method[26]. Then, in an initial breakthrough, the reaction of insoluble **L-Au₂** with 1,4-bis(diphenylphosphino)butane (dppb) led to formation of single crystals of the remarkable $3_1$ knot complex [$Au_6${1,2-$C_6H_4(OCH_2CC)_2$}₃{$Ph_2P(CH_2)_4PPh_2$}₃], **Au₆** (Fig. 1). This complex **Au₆** can be considered as a trimer of the cyclic digold(I) complex [$Au_2${1,2-$C_6H_4(OCH_2CC)_2$}{$Ph_2P(CH_2)_4PPh_2$}], **Au₂** (Fig. 1)[27].

The complex **Au₆** was crystallized in two ways, by vapor diffusion of ether into the filtered reaction solution. From tetrahydrofuran (thf) at room temperature, crystals formed in the space group *P-1* with the unit composition 2[**Au₆**].3thf.2H₂O, with two independent but similar chiral molecules of **Au₆** in the unit cell. From dichloromethane at 4 °C, crystals formed in the space group *P2₁/c* with the unit composition [**Au₆**].3CH₂Cl₂. In each case, the inversion symmetry of the space group generates the respective enantiomers. There was no crystallographically imposed symmetry of the individual molecules but each has the approximate symmetry of the ideal trefoil knot, which has the $D_3$ point group symmetry. As depicted in Figs. 1 and 2, the $Au_6$ trefoil knot was constructed of six Au atoms, three diacetylide and three

dppb ligands, and each gold(I) atom has roughly linear C–Au–P stereochemistry, with range of distances C–Au 1.980(3)–2.050(3) Å and P–Au 2.2699(3)–2.2899(4) Å. The three L ligands are arranged around the outer edges, with roughly 45° angles between the approximate $C_3$ axis and the ligand panels, determining the overall size of the knot, which is ~1.07 nm from the center to the outmost atoms and ~0.7 nm in thickness. The flexible methylene chains of the dppb ligands occupy much of the inner space of the knot, so allowing formation of the very compact structure (Fig. 2, Supplementary Figs. 3–7). Most notably, the number of backbone atoms forming this $Au_6$ knot is 54 including 6 Au atoms (BCR = 18) much smaller than the previous reported smallest organic trefoil knot with a 76 atom backbone (BCR = 25.3) or metallaknot with a 69 atom backbone (BCR = 23)[12,18] and close to the predicted value of 50 for the smallest poly(methylene) trefoil knot[21]. The BCR value of 18 for the **Au₆** knot is also lower than the previous record low value of BCR = 24 for the $8_{19}$ organic knot or 23 for the smallest trefoil metallaknot[19], indicating that **Au₆** is not only the smallest and tightest trefoil knot but also the tightest of all known molecular knots[6,9]. Compared to the potential poly(methylene) trefoil knot[16], the low steric requirement of the gold-acetylide units may be expected to favor knot formation, while the rigidity of these linear units will be unfavorable, and the bulky diphenylphosphino units clearly cannot be part of the knot interior.

The above results prove the presence of the unique $Au_6$ knot in the crystalline state, and it was of interest to investigate its spectroscopic properties and stability in solution. Previous studies have shown the Au–P bonds in related diacetylide-diphosphine complexes are labile and that this allows rapid interconversion between isomers, so that equilibrium mixtures often exist in solution. Thus, the simple $Au_2$ ring may equilibrate with the 2[catenane] or the double $Au_4$ ring (Fig. 3)[22]. The most complex reported example is the Solomon link, which formed on crystallization but dissociated back to the simple ring complex in solution[22]. The formation of the 2[catenane] or Solomon link is favored by the formation of aurophilic bonds (Fig. 3), but the

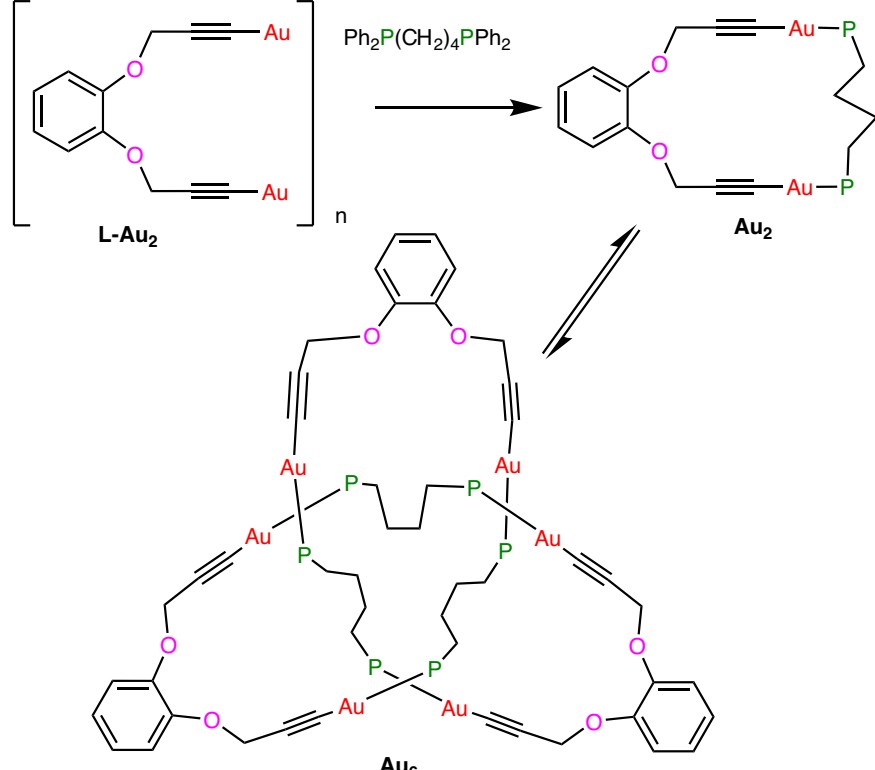

**Fig. 1 | Preparation of a molecular $3_1$ golden knot Au₆.** This is the smallest molecular knot characterized to date. Phenyl groups on the diphosphine ligand are omitted, for clarity.

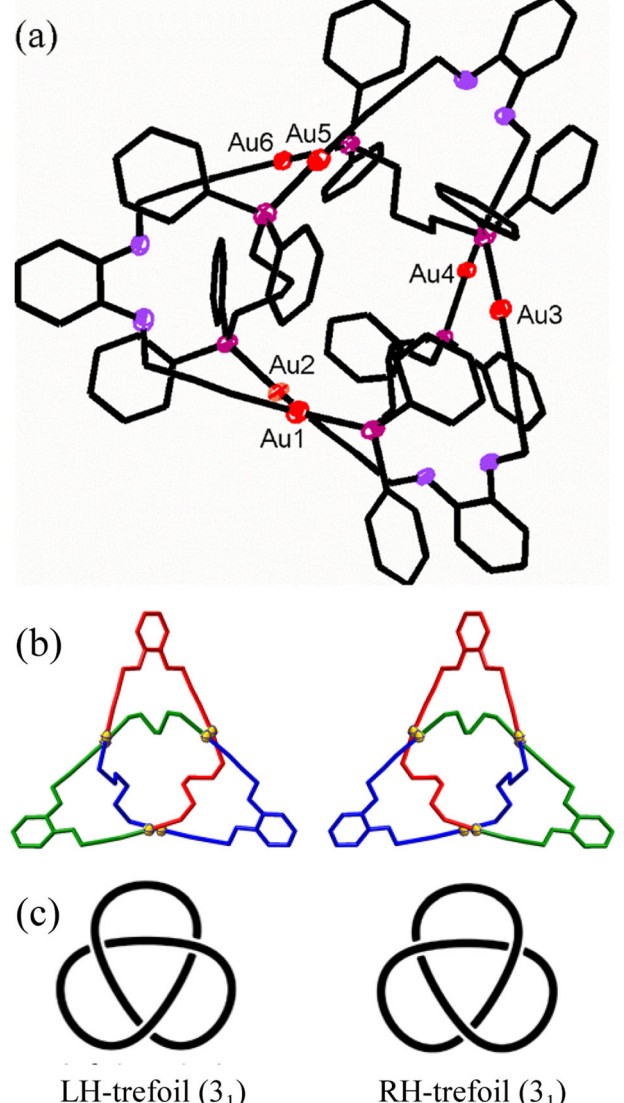

**Fig. 2 | The structure of the trefoil metallaknot [Au₆(L)₃(dppb)₃], Au₆.** Separate sections are **a** the structure (color code: Au, red; P, purple; O, mauve); **b** the core structures of enantiomers (H atoms and dppb phenyl groups omitted for clarity); **c** the ideal left- and right-handed trefoil knots.

structure of **Au₆** indicates that aurophilic bonding is either absent or weak (Fig. 2). Thus, the shortest Au···Au distances in the thf and CH₂Cl₂ solvates are Au···Au = 3.68 Å and 3.44 Å, respectively, and the full range of Au···Au distances is 3.44−5.44 Å, with only the shortest one in the accepted range of 2.85−3.50 Å for aurophilic bonds[28]. Many of the gold atoms have CH···Au interactions in the range 3.0−3.1 Å, which could be very weakly bonding (Supplementary Figs. 5−7). Of course, entropy strongly favors the smallest molecular unit, namely the **Au₂** ring.

The $^{31}$P NMR spectrum, obtained by dissolving crystals of **Au₆** in CD₂Cl₂, contained a singlet resonance at δ 39.2, significantly shifted from the value [Au₂Cl₂(dppb)] (δ 29.6) but similar to values found in analogs of **Au₂**[29,30]. However, a single $^{31}$P resonance is expected for each of the **Au₂** and **Au₆** complexes due to their symmetry (Fig. 1), so this does not distinguish between the two possible structures. The $^{1}$H NMR spectrum is more informative and is shown in Fig. 4 (it is unchanged at −20 °C). For **Au₂** with effective $C_{2v}$ symmetry, the protons in each CH₂ group and the phenyl groups in each PPh₂ unit should be equivalent but, for the chiral complex **Au₆** with effective $D_3$ symmetry, they are diastereotopic and the tightness of the knot should

cause a clear distinction[6,9]. No distinction was observed (Fig. 4). The dppb resonances are broad with unresolved couplings but the CH₂O resonance occurs as a sharp singlet at 4.78 ppm, which is similar to the value for the free ligand (Fig. 4) and as expected for the ring **Au₂**, and not as the AB multiplet expected for the CH$^A$H$^B$O protons of chiral **Au₆**. This provides strong evidence that the trefoil knot **Au₆** dissociates in solution to the simple ring complex **Au₂**, in a similar way as for the Solomon link complex (Fig. 3)[29,30]. The ESI-MS in acetonitrile solution supports this conclusion.

## Discussion

The experimental work indicates that the **Au₂** ring complex is more stable than the knot complex **Au₆** in solution in dichloromethane, and the structure determinations (Supplementary Figs. 5−7) identify no very strong attractive secondary bonding interactions that might balance the unfavorable entropic change on trimerization of **Au₂** to form **Au₆**. DFT calculations (Supplementary Fig. 8) predict that trimerization of **Au₂** to form **Au₆** is unfavorable in solution, and that the knot complex **Au₆** is also higher in energy than the corresponding open Au₆ ring. The formation by self-assembly of the unique knot complex **Au₆** was therefore unpredictable. One possible explanation is that the packing of the compact **Au₆** knot is more efficient than for the open ring compounds and that the knot forms only at the surface of the growing crystal lattice, in an analogous way as for some coordination polymers[31,32].

In summary, the first σ-bonded molecular knot with gold(I) centers in the backbone, **Au₆** (Figs. 1, 2), was prepared by a simple self-assembly procedure and its remarkable structure was determined by X-ray crystallography (Fig. 2). Of note, the trefoil knot **Au₆** contains only 54 atoms in the backbone and hence it is the smallest molecular knot yet reported, while its backbone crossing ratio of 18 also defines it as the tightest molecular knot[6,9]. Since recent discoveries have indicated a strong dependence of knot tightness on useful properties, this work should provide a strong motivation to pursue similar, but hopefully more robust, structures by self-assembly[6,12,16,17,31–33].

## Methods

UV-visible spectra were measured using a Shimadzu UV-1800 spectrophotometer in CH₂Cl₂ solution. MALDI-TOF-MS was performed using an ABI MALDI TOF/TOF 5800 in a positive ion mode using trans-2-[3-(4-tert-butylphenyl)-2-methyl-2-propenyldidene]malononitrile as matrix material. ESI-MS was performed by using an electrospray PE-Sciex mass spectrometer as a solution in acetonitrile with NaOTf to aid ionization. IR spectra were recorded using a Bruker vertex 70 infra-red spectrometer (resolution 5 cm$^{-1}$, scan: 32, scale: 600−4000 cm$^{-1}$) as KBr disc. NMR spectra were recorded using an AVANCE III 400 MHz (Bruker) spectrometer. The DFT calculations were carried out using the BLYP functional, with double-zeta basis set and first-order scalar relativistic corrections. In each case, several potential conformers were tested first by molecular mechanics minimization followed by DFT, with the lowest energy reported. The solvent effect of dichloromethane was modeled by using COSMO, all as implemented in ADF-2020[34]. Relative Gibbs free energies are given in Supplementary Fig. 8.

For structure determinations, a suitable crystal was mounted on a XtaLAB AFC11 (RCD3) diffractometer (thf solvate) or Bruker APEX-II CCD diffractometer (CH₂Cl₂ solvate). Data reduction, cell refinement, and experimental absorption correction were performed with the software package of CrysAlisPro. The structures were solved by intrinsic phasing methods by SHELXT 2018 and refined against F$^2$ by full-matrix least-squares by SHELXL 2018[35,36]. All non-hydrogen atoms were refined anisotropically. Hydrogen atoms were generated geometrically. All calculations were carried out by the program package of Olex2 (ver 1.2.10.32)[37]. Electron density due to disordered solvent molecules in the thf solvate was treated by using SQUEEZE[38]. Crystallographic data are given in Supplementary Tables 1 and 2 and in

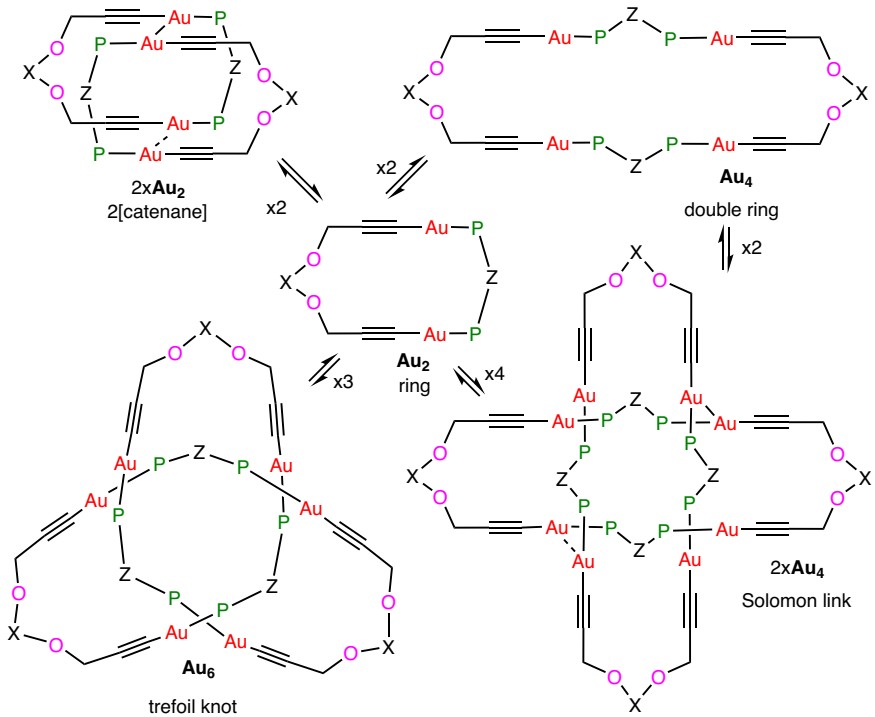

**Fig. 3 | Self-assembly motifs from a simple Au₂ ring.** Several [2]catenanes are known but only one trefoil knot and one Solomon link. Abbreviations used are P = PPh₂, X, Z = linker groups.

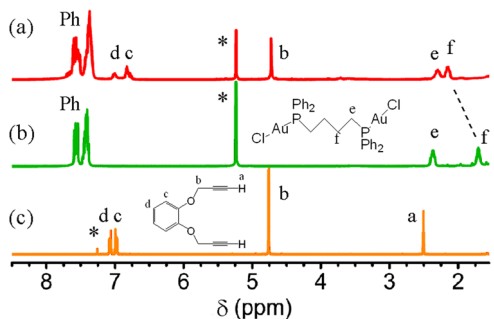

**Fig. 4 | ¹H NMR spectra (400 MHz) of complexes and ligand.** Separate sections are **a** solution obtained by dissolving the **Au₆** knot in CD₂Cl₂; **b** [Au₂Cl₂(Ph₂P(CH₂)₄PPh₂)] in CD₂Cl₂; **c** H₂L in CDCl₃. The peak marked * is due to CHDCl₂ or CHCl₃.

the CIF file, which has been deposited at the Cambridge Crystallographic Data Centre, under deposition number CCDC 2278951. Copies of the data can be obtained free of charge via https://www.ccdc.cam.ac.uk/structures/.

### Synthesis of 1,2-C₆H₄(OCH₂CCH)₂, LH2

To a solution of 1,2-dihydroxybenzene (18.6 g, 0.10 mol) in acetone (250 mL) was added propargyl bromide (26.2 g, 0.22 mol) and Cs₂CO₃ (97 g, 0.3 mol). The solution was refluxed for 16 h. After evaporation of the solvent, the crude product was purified by column chromatography using silica gel and chloroform as eluant. It was isolated as a white powder. Yield 86%. NMR in CDCl₃: δ(¹H) = 7.08 (m, 2H), 6.98 (m, 2H), 4.77 (s, 4H, CH₂), 2.51 (s, 2H, CH); δ(¹³C) = 147.8 (C), 122.2 (CH), 115.1 (CH), 78.7 (CH₂), 76.2 (C), 57.0 (CH).

#### Synthesis of 1,2-C₆H₄(OCH₂CCAu)₂, LAu₂.

To a suspension of [AuCl(SMe₂)] (0.52 g, 1.77 mmol) in tetrahydrofuran (50 mL) was added a solution of **LH2** (0.17 g, 0.89 mmol) and sodium acetate (0.15 g, 0.9 mmol) in methanol (15 mL). The

mixture was stirred at room temperature for 3 h, to yield a yellow precipitate, which was separated by filtration, washed with tetrahydrofuran and then pentane, and dried under vacuum. Yield 0.40 g, 78%. The complex is insoluble in common organic solvents. IR(Nujol): n(C≡C) = 2000 cm⁻¹ (w).

### Synthesis of Au₆ knot

Solvate 2**Au₆**.2H₂O.3thf. The L-Au₂ (10 mg, 0.017 mmol) was dispersed in thf (2 mL), followed by the addition of dppb (dppb = 1,4-bis(diphenylphosphino)butane) (7 mg) under rapid stirring. After 12 h, residual insoluble solid was removed by filtration, followed by vapor diffusion of diethyl ether into the filtrate to give colorless crystals of 2**Au₆**.2H₂O.3thf. Yield: 7.2 mg, 42%. MALDI-TOF-MS: *m/z* = 623.8 [Au(dppb)]⁺, 1005.0 [Au₂L(dppb)]H⁺, 1201.0 [Au₃L(dppb)]⁺. IR: ν(C≡C) = 2142 cm⁻¹. NMR in CD₂Cl₂ (**Au₂**): δ(¹H) = 2.10 (m, 4H, CH₂C), 2.23 (m, 4H, CH₂P), 4.78 (s, 4H, CH₂O), 6.75 (m, 2H, C3,C6, catechol), 7.05 (m, 2H, C4,C5, catechol), 7.3 (m, 12H, H*o*,H*p*, PhP), 7.6 (m, 8H, H*m*, PhP); δ(³¹P) = 39.2 (s).

Solvate **Au₆**.3CH₂Cl₂. The L-Au₂ (10 mg, 0.017 mmol) was dispersed in CH₂Cl₂ (2 mL) followed by the addition of dppb (dppb = 1,4-bis(diphenylphosphino)butane) (7 mg) dissolved in CH₂Cl₂ (2 mL) dropwise under rapid stirring. After 12 h at room temperature, the residual solid was removed by filtration. Colorless crystals of **Au₆**.3CH₂Cl₂ were obtained by the diffusion of diethyl ether into the filtrate for two weeks at 4 °C in a refrigerator. Yield: 3.5 mg, 18%. ESI-MS (MeCN/NaOTf): Calc. for Au₂(L)(dppb)Na⁺, *m/z* = 1027.14; Found, *m/z* = 1027.20. Anal. Calc. for C₁₂₃H₁₁₄Au₆Cl₆O₆P₆: C 45.20, H 3.52. Found: C 45.31, H 3.80%.

### Data availability

The online version of this article contains supplementary information. The X-ray crystallographic coordinates for structures reported in this study have been deposited at the Cambridge Crystallographic Data Centre (CCDC), under deposition number CCDC2278951. These data can be obtained free of charge from The Cambridge Crystallographic Data Centre via www.ccdc.cam.ac.uk/data_request/cif. All other data are available from the corresponding authors upon request.

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

## Acknowledgements

We acknowledge the financial support by Liaoning Natural Science Foundation of China (2020-MS-024 to G.L.) and discussions with Prof. Weibin Yu.

## Author contributions

Z.L. and J.Z. made equal contributions to the experimental research, G.L. and R.J.P. designed the research, and all authors contributed to data analysis and writing the manuscript.

## Competing interests

The authors declare no competing interests.
