## [Peer Review File · Nature Communications]

Self-assembly of the smallest and tightest molecular knot: A hexagold(I) trefoil metallaknot with chain length of only 54 atomsReviewers' Comments:

Reviewer #1:

Remarks to the Author:

This is a really nice manuscript that describes the serendipitous observation that a digold macrocycle that is in dynamic exchange with other isomers through metal ligand exchange crystallises unexpectedly as a trefoil knot. The formation of the knot is demonstrated crystallographically (please note that I cannot review the quality of the crystallography - an expert opinion should be sought as the manuscript rests on this data).

The observation of the knot is surprising and definitely suitable for publication in Nat Commun - there has recently been great interest in molecular knots and the physical limits in terms of chain length/tightness and this article contributes to that discussion. I have a couple of questions/points the authors should address prior to publication:

1. The authors point out that the aurophilic interactions that one might expect in the knot are absent. However, they don't really comment on what interactions are observed in the crystal structure - a brief discussion should be added. They should also increase the size of the structure in Fig 2 - given that this is the most important figure in the paper it should be easier to read. They might want to consider adding zoomed in regions that are relevant to the discussion.

2. In the discussion of the limits of knot tightness a key factor seems to be overlooked. As far as I can tell, the theoretical studies have focused on structures in which there are no (apart from dispersion) attractive interactions between the chains at the crossing points. It seems perfectly reasonable that attractive non covalent interactions would increase the stabilization of tight knots. This point should be made somewhere and related to the discussion suggested in (1).

3. The solution phase NMR study is interesting but feels incomplete. Perhaps it lies beyond the current report but it seems logical to perform the same study in different solvents and at different temperatures. I completely accept the authors' argument that the trefoil seems to be stabilized in the solid state wrt to the macrocycle. However, other solvents or lower T may also favor this in solution and it must be part of the combinatorial library if it crystallises...

4. Some typos/points for clarification:

"Hence, how small a molecular knot could remain an open question." - A word is missing here?

"The most complex example is the Solomon link which formed on crystallization but dissociated back to the simple ring complex in solution.¹⁷" It should be made clearer that this is a previous result - when I read this first I thought that they had observed the Solomon link in this case.

" However, the ³¹P atoms in both Au₂ and Au₆ (Figure 1) are expected to be equivalent so this does not distinguish between the two possible structures." - The current wording could be misunderstood to mean that the P atoms are equivalent in the Au₂ and Au₆ species. Perhaps "A single resonance is expected for the each of the Au₂ and Au₆ species due to their symmetry"?

"AB" doesn't need to be in quote marks, it is accepted nomenclature.

5. ESI - generally lacking in detail.

Calculations: no details are provided for how the calculations were performed (program, model, basis set, solvation, whether the values are Gibbs or electronic). This has to be improved dramatically. I would expect to see details of how the input structure was prepared and all computational details required to reproduce the same result. At this point it is hard to judge if the computational data is of

sufficient rigor/accuracy to actually be useful for the discussion.

MS: I'm not sure what is going on in Fig S3 - was an m/z value that corresponds to Au₆ observed? If not, what is the purpose of the figure? It would also be useful to know how the sample was prepared - if from solution then it is not surprising that no Au₆ ion is present.

Reviewer #2:

Remarks to the Author:

This paper by Li, Puddephat and co-workers report on the discovery of the smallest and tightest molecular knot to date. A trefoil (31) metalloknot with 6 gold atoms in its backbone was found to have crystallized out of a dynamic mixture of gold-phosphine ligand complexes, and was isolated in around 40% yield with respect to the starting ligands. The knot only has 54 atoms in its backbone, (much shorter than the previously tightest knot with 76 atoms, ref 13) and its so-called backbone-to-crossing ratio of 18 is approaching the theoretically lowest predicted limit for molecular trefoil knots. Knot tightness is the most relevant parameter for predicting emergent properties stemming from entanglements, so the discovery that knots as extremely tight as this can emerge spontaneously and from a dynamic mixture is very unexpected and cool.

I found the crystal structure of the trefoil knot absolutely spectacular and the discovery is a beautiful result which is definitely worthy of publication in Nat. Commun. This structure will, I believe, strongly change how people think about knot design and knot tightness.

However, I think the rest of the paper is lacking and feels a bit prematurely submitted. I think both discussion, referencing and experimental section needs to be updated, and some minor additional experiments added. In principle I am very supportive of publication, but I think at the level of top generalist journals like Nat. Commun. the paper/presentation needs to match the quality of the scientific discovery. I hence urge the authors to consider the following changes, after which I would be (strongly) supportive of publication:

* The discussion of tightness in the introduction relies heavily on a paragraph in the Dave Leigh review (ref 6), but it should also be mentioned that Leigh only compares organic knots and not metalloknots. Introducing metals and bigger atoms such as P in the backbone of a knot instead of C/N/O obviously changes bond distances and angles that can be obtained to a large degree, so I think this needs to be mentioned and discussed adequately in the introduction and conclusions. I would also urge the authors to compare their BCRs to other metalloknots, for example from the Jin and Fujita groups (see below), as this would yield a better comparison than the table in Ref 6.

* The referencing in the paper needs significant work. The manuscript in its current state is underreferenced and omits many key papers and in particular pretty much all recent discoveries. First of all, there have now been several papers probing the practical effects of knot tightness, for example PNAS, 2019, 2452–2457; Chem. Sci., 2021, 1826; CEJ, 2020, 1576; Nature, 2020, 562; ACIE, 2019, 11324. Some or all of these should be cited in the intro. I also lack a discussion on the usefulness of molecular knots that would better inform readers why the discovery in this paper is so important. Key papers to cite here could for example be Science, 2016, 1555; ACIE, 2018, 10484; Nat. Chem. 2020, 939; JACS, 2020, 18859; Chem. Sci., 2019, 5884. The single biggest oversight is however the omission of essentially all recent metalloknot papers, a blossoming field with strong progress in recent years. In particular, the recent Chem. Rev. from the Jin group on metalloknots is a major omission, and this paper definitely needs to be cited (Chem. Rev., 2020, 6288). The extensive recent work from Jin (i.e. JACS, 2021, 1119; JACS, 2020, 18946; JACS, 2023, 4746; JACS, 2023, 725 etc), Fujita (i.e. Nat. Commun., 2019, 921; Chem, 2020, 294; JACS, 2021, 16734, etc) and Chi (Angew. Chem., Int. Ed., 2018, 57, 5669; JACS, 2020, 9327) should also be acknowledged to some extent.

* In terms of experimental work, I find that the authors should maybe try a few additional simple experiments before the story is finished. The current discussion is very limited and mainly speculative. It is clear from the NMR analysis that the system is dynamic to some extent, but it would be good with DOSY NMR measurements to determine the size and quantity of the species in solution as well.

Furthermore, mass spectrometry from the solution state would be informative and should be performed. Also, the MALDI results in the SI are not in full agreement with the structural characterization of the knot. I'm not doubting the crystal data (the knot is clearly there!), but as MALDI often fragments weak bonds like these I was just wondering how the authors prepared their MALDI samples and if altered sample prep could help them catch mass of the full knot.

* It is clear the knot is not the major form in solution, but then the authors use calculations to claim it is also less stable than the corresponding macrocycle. These calculations need to be better described, and in much more detail. Is this gas phase for example? How is energy minimized? There are some scattered sentences on this in the general experimental but this is too vague to understand what has been done. It is interesting that the knot would be disfavored over the macrocycle as, in such a case, crystallization of the trefoil knot must be a very strong kinetic driving force. The authors need to discuss this in more detail and try to devise a (proper!) model for why the knot forms. Can you do seeding experiments with the already obtained crystals for example? Can you use prediction software to compare packing energies for macrocycle and knot?

Finally, some additional thoughts:

*The title is not very scientific, and could be rewritten to sound a bit less colloquial.

*Yield is only given for the THF solvate, and not the DCM solvate.

*Phrase "several billion primary knots of increasing complexity": Maybe revise, as the amount of primary knots is predicted to be infinite.

* Phrase "In contrast to conventional knots, this tight molecular knot is also easily untied!". This is not true, a majority of reported knots are dynamic in nature (either via dynamic covalent bonds or metal-ligand interactions) and disassembles readily. This is also generally a drawback, not an advantage. Revise?

* Phrase "when the structure is projected onto a two-dimensional surface": Specify that it must be the reduced form of the structure.

Reviewer #3:

Remarks to the Author:

The manuscript submitted by Puddephatt et al. describes the self-assembly of a tight golden trefoil knot following an approach well established in the group, which had previously yielded singly and doubly interlocked catenanes (see ACIE 1999, 38, 3376 and ACIE 2000, 39, 3819). The current report is exclusively based on a solid-state structure that unambiguously demonstrates the knotted structure of the assembly. This is a beautiful structure that deserves publication in Nature Communications. Sadly, NMR spectroscopy shows the trefoil knot does not exist in solution, so no further solution-based experiment was carried out. Yet, the simplicity of this knot synthesis (only 54 atoms) makes this piece of work particularly appealing.

The methodology is sound and the data well interpreted. All claims are supported by the appropriate evidences.

REVIEWER COMMENTS

Reviewer #1 (Remarks to the Author):

This is a really nice manuscript that describes the serendipitous observation that a digold macrocycle that is in dynamic exchange with other isomers through metal ligand exchange crystallises unexpectedly as a trefoil knot. The formation of the knot is demonstrated crystallographically (please note that I cannot review the quality of the crystallography - an expert opinion should be sought as the manuscript rests on this data).

The observation of the knot is surprising and definitely suitable for publication in Nat Commun - there has recently been great interest in molecular knots and the physical limits in terms of chain length/tightness and this article contributes to that discussion. I have a couple of questions/points the authors should address prior to publication:

1. The authors point out that the aurophilic interactions that one might expect in the knot are absent. However, they don't really comment on what interactions are observed in the crystal structure - a brief discussion should be added. They should also increase the size of the structure in Fig 2 - given that this is the most important figure in the paper it should be easier to read. They might want to consider adding zoomed in regions that are relevant to the discussion.

Response: The discussion on secondary bonds is modified, as requested, and new Figures S5-S7 are added to the SI to show details. The size of the structure in Fig 2 is also increased so that the trefoil knot structure is clearer, as requested. The conclusion remains – that the secondary bonding in the knot structure is too weak to favour formation of the knot in solution.

2. In the discussion of the limits of knot tightness a key factor seems to be overlooked. As far as I can tell, the theoretical studies have focused on structures in which there are no (apart from dispersion) attractive interactions between the chains at the crossing points. It seems perfectly reasonable that attractive non covalent interactions would increase the stabilization of tight knots. This point should be made somewhere and related to the discussion suggested in (1).

Response: This is a good point and the discussion is modified to outline the pros and cons of methylene versus gold acetylide units for knot formation.

3. The solution phase NMR study is interesting but feels incomplete. Perhaps it lies beyond the current report but it seems logical to perform the same study in different solvents and at different temperatures. I completely accept the authors' argument that the trefoil seems to be stabilized in the solid state wrt to the macrocycle. However, other solvents or lower T may also favor this in solution and it must be part of the combinatorial library if it crystallises...

Response: This is a good suggestion. The NMR in CD₂Cl₂ is unchanged on cooling to -20°C (now noted in the text). The ability to study solvent effects is limited by the low solubility of

Au₆ – for example it is very sparingly soluble in acetone or benzene. Our earlier studies on related compounds have shown that equilibria in solution are established in minutes at room temperature, but more slowly than the time scale for NMR fluxionality. We have not found conditions where higher oligomer than Au₂ is present in significant quantity. Reviewer 2 made a related comment.

4. *Some typos/points for clarification:*

"Hence, how small a molecular knot could remain an open question." - A word is missing here?

Response: Yes, this was an error – now corrected.

"The most complex example is the Solomon link which formed on crystallization but dissociated back to the simple ring complex in solution.¹⁷" It should be made clearer that this is a previous result - when I read this first I thought that they had observed the Solomon link in this case.

Response: yes – it is now made clearer that this complex was previously reported.

" However, the 3IP atoms in both Au₂ and Au₆ (Figure 1) are expected to be equivalent so this does not distinguish between the two possible structures." - The current wording could be misunderstood to mean that the P atoms are equivalent in the Au₂ and Au₆ species. Perhaps "A single resonance is expected for the each of the Au₂ and Au₆ species due to their symmetry"?

Response: I agree this is better wording, and the change is made.

"AB" doesn't need to be in quote marks, it is accepted nomenclature.

Response: The quote marks are removed, as requested.

5. *ESI - generally lacking in detail.*

Calculations: no details are provided for how the calculations were performed (program, model, basis set, solvation, whether the values are Gibbs or electronic). This has to be improved dramatically. I would expect to see details of how the input structure was prepared and all computational details required to reproduce the same result. At this point it is hard to judge if the computational data is of sufficient rigor/accuracy to actually be useful for the discussion.

Response: Most of the general details were given in the Methods section, but the solvent correction was omitted and is now included. The .xyz file gives details of the calculated structures. A more sophisticated theoretical study would be valuable but is beyond our expertise (see related comment by reviewer 2).

MS: I'm not sure what is going on in Fig S3 - was an m/z value that corresponds to Au₆ observed? If not, what is the purpose of the figure? It would also be useful to know how the sample was prepared - if from solution then it is not surprising that no Au₆ ion is present.

Response: The MS was recorded by MALDI-TOF using trans-2-[3-(4-tert-butylphenyl)-2-methyl-2-propenyldiene]malononitrile as matrix material, deposited by evaporation from CH₂Cl₂. I have to agree that the result is more likely to confuse than enlighten, so the figure is removed, as suggested.

Reviewer #2 (Remarks to the Author):

This paper by Li, Puddephat and co-workers report on the discovery of the smallest and tightest molecular knot to date. A trefoil (31) metalloc knot with 6 gold atoms in its backbone was found to have crystallized out of a dynamic mixture of gold-phosphine ligand complexes, and was isolated in around 40% yield with respect to the starting ligands. The knot only has 54 atoms in its backbone, (much shorter than the previously tightest knot with 76 atoms, ref 13) and its so-called backbone-to-crossing ratio of 18 is approaching the theoretically lowest predicted limit for molecular trefoil knots. Knot tightness is the most relevant parameter for predicting emergent properties stemming from entanglements, so the discovery that knots as extremely tight as this can emerge spontaneously and from a dynamic mixture is very unexpected and cool.

I found the crystal structure of the trefoil knot absolutely spectacular and the discovery is a beautiful result which is definitely worthy of publication in Nat. Commun. This structure will, I believe, strongly change how people think about knot design and knot tightness.

However, I think the rest of the paper is lacking and feels a bit prematurely submitted. I think both discussion, referencing and experimental section needs to be updated, and some minor additional experiments added. In principle I am very supportive of publication, but I think at the level of top generalist journals like Nat. Commun. the paper/presentation needs to match the quality of the scientific discovery. I hence urge the authors to consider the following changes, after which I would be (strongly) supportive of publication:

** The discussion of tightness in the introduction relies heavily on a paragraph in the Dave Leigh review (ref 6), but it should also be mentioned that Leigh only compares organic knots and not metalloc knots. Introducing metals and bigger atoms such as P in the backbone of a knot instead of C/N/O obviously changes bond distances and angles that can be obtained to a large degree, so I think this needs to be mentioned and discussed adequately in the introduction and conclusions. I would also urge the authors to compare their BCRs to other metalloc knots, for example from the Jin and Fujita groups (see below), as this would yield a better comparison than the table in Ref 6.*

Response: This is a good point. I have modified the discussion when comparing this system to the poly(methylene) one. A related point was made by Reviewer 1. I have also added references to the pioneering work of the Jin/Fujita groups, as requested. I agree that better acknowledgment of the metalloc knot field was needed, though I did not find concerted effort yet to minimise size or maximise tightness in the metalloc knot field. Perhaps there should be.

** The referencing in the paper needs significant work. The manuscript in its current state is underreferenced and omits many key papers and in particular pretty much all recent discoveries. First of all, there have now been several papers probing the practical effects of knot tightness, for example PNAS, 2019, 2452–2457; Chem. Sci., 2021, 1826; CEJ, 2020, 1576; Nature, 2020,*

562; *ACIE*, 2019, 11324. Some or all of these should be cited in the intro. I also lack a discussion on the usefulness of molecular knots that would better inform readers why the discovery in this paper is so important. Key papers to cite here could for example be *Science*, 2016, 1555; *ACIE*, 2018, 10484; *Nat. Chem.* 2020, 939; *JACS*, 2020, 18859; *Chem. Sci.*, 2019, 5884. The single biggest oversight is however the omission of essentially all recent metalloknot papers, a blossoming field with strong progress in recent years. In particular, the recent *Chem. Rev.* from the Jin group on metalloknots is a major omission, and this paper definitely needs to be cited (*Chem. Rev.*, 2020, 6288). The extensive recent work from Jin (i.e. *JACS*, 2021, 1119; *JACS*, 2020, 18946; *JACS*, 2023, 4746; *JACS*, 2023, 725 etc), Fujita (i.e. *Nat. Commun.*, 2019, 921; *Chem*, 2020, 294; *JACS*, 2021, 16734, etc) and Chi (*Angew. Chem., Int. Ed.*, 2018, 57, 5669; *JACS*, 2020, 9327) should also be acknowledged to some extent.

Response: Again this is a valid point and I have included a selection of the recommended papers. I think to include them all in a short communication would be unjustified, so I have tried to pick at least one most relevant article from each major player relevant to knot tightness or metallaknots.

** In terms of experimental work, I find that the authors should maybe try a few additional simple experiments before the story is finished. The current discussion is very limited and mainly speculative. It is clear from the NMR analysis that the system is dynamic to some extent, but it would be good with DOSY NMR measurements to determine the size and quantity of the species in solution as well. Furthermore, mass spectrometry from the solution state would be informative and should be performed. Also, the MALDI results in the SI are not in full agreement with the structural characterization of the knot. I'm not doubting the crystal data (the knot is clearly there!), but as MALDI often fragments weak bonds like these I was just wondering how the authors prepared their MALDI samples and if altered sample prep could help them catch mass of the full knot.*

Response: The conditions for the MALDI-MS are now given in more detail, as requested. The DOSY NMR experiment is most valuable when different oligomers are present in solution. Unfortunately, we could not find conditions when the knot (or other oligomer) was present in detectable amounts in solution. Additionally, the DOSY diffusion rates for open rings and catenane dimers are found to be similar (ref 29), so it is unlikely to give very useful information. See also response to related points from reviewer 1.

** It is clear the knot is not the major form in solution, but then the authors use calculations to claim it is also less stable than the corresponding macrocycle. These calculations need to be better described, and in much more detail. Is this gas phase for example? How is energy minimized? There are some scattered sentences on this in the general experimental but this is too vague to understand what has been done. It is interesting that the knot would be disfavored over the macrocycle as, in such a case, crystallization of the trefoil knot must be a very strong kinetic driving force. The authors need to discuss this in more detail and try to devise a (proper!) model for why the knot forms. Can you do seeding experiments with the already obtained crystals for example? Can you use prediction software to compare packing energies for macrocycle and knot?*

Response: More details are now given on the approach used for calculations, as requested. We do not have the software or expertise to calculate relative packing energies. I agree completely that it would be good to have a more convincing rationale for knot formation. Our suggestion that the lattice energy is greater for the very compact knot structure than for the floppy ring structures is, I think, most probable. But an advanced theory approach would certainly be most valuable.

Finally, some additional thoughts:

**The title is not very scientific, and could be rewritten to sound a bit less colloquial.*

Response: A new title is suggested, as requested.

**Yield is only given for the THF solvate, and not the DCM solvate.*

Response: More characterization data now given, as requested.

**Phrase “several billion primary knots of increasing complexity”: Maybe revise, as the amount of primary knots is predicted to be infinite.*

Response: Yes, the sentence is now rephrased.

** Phrase “In contrast to conventional knots, this tight molecular knot is also easily untied!”. This is not true, a majority of reported knots are dynamic in nature (either via dynamic covalent bonds or metal-ligand interactions) and disassembles readily. This is also generally a drawback, not an advantage. Revise?*

Response: This was a somewhat flippant comment relating to the knots that children make with string, but it was inappropriate and is now removed.

** Phrase “when the structure is projected onto a two-dimensional surface”: Specify that it must be the reduced form of the structure.*

Response: Yes, the change is made, as requested

Reviewer #3 (Remarks to the Author):

The manuscript submitted by Puddephatt et al. describes the self-assembly of a tight golden trefoil knot following an approach well established in the group, which had previously yielded singly and doubly interlocked catenanes (see ACIE 1999, 38, 3376 and ACIE 2000, 39, 3819). The current report is exclusively based on a solid-state structure that unambiguously demonstrates the knotted structure of the assembly. This is a beautiful structure that deserves publication in Nature Communications. Sadly, NMR spectroscopy shows the trefoil knot does not exist in solution, so no further solution-based experiment was carried out. Yet, the simplicity of this knot synthesis (only 54 atoms) makes this piece of work particularly appealing.

The methodology is sound and the data well interpreted. All claims are supported by the appropriate evidences.

Response: No changes recommended.

Reviewers' Comments:

Reviewer #1:

Remarks to the Author:

I'd like to thank the authors for addressing my comments and those of the other reviewers. I recommend the manuscript is accepted in its current form.

Reviewer #2:

Remarks to the Author:

The authors have responded thoroughly to the comments by myself and the other referees, and I think the paper is stronger now. This study is fundamentally very simple, but after the revisions it also has everything there for a complete story. It would in my view still be nice with a better explanation for why the knot forms (for example via theoretical approaches), but this is not a hill I have to die on if the editor and other referees agree to proceed to publication with the current manuscript.

Reviewer #3:

Remarks to the Author:

The authors have addressed all the comments, if we consider that additional solution-state experiments are not possible. The changes made to Figure 2 do not improve its readability, in my opinion. Perhaps the use of colours could help?

REVIEWERS' COMMENTS

Reviewer #1 (Remarks to the Author):

I'd like to thank the authors for addressing my comments and those of the other reviewers. I recommend the manuscript is accepted in its current form.

Response: No further changes recommended.

Reviewer #2 (Remarks to the Author):

The authors have responded thoroughly to the comments by myself and the other referees, and I think the paper is stronger now. This study is fundamentally very simple, but after the revisions it also has everything there for a complete story. It would in my view still be nice with a better explanation for why the knot forms (for example via theoretical approaches), but this is not a hill I have to die on if the editor and other referees agree to proceed to publication with the current manuscript.

Response: I agree that it would be good to have a better mechanistic understanding, but our best efforts have been unsuccessful.

Reviewer #3 (Remarks to the Author):

The authors have addressed all the comments, if we consider that additional solution-state experiments are not possible. The changes made to Figure 2 do not improve its readability, in my opinion. Perhaps the use of colours could help?

Response: The many overlaps do make the structure complex but I think readers should be able to follow it with the help of the simplified figures below the main one and the ChemDraw figures.